# Empirical Investigation of How Social Media Usage Enhances Employee Creativity: The Role of Knowledge Management Behavior

**DOI:** 10.3390/bs13070601

**Published:** 2023-07-18

**Authors:** Huiqin Zhang, Meng Wang, Anhang Chen

**Affiliations:** College of Management Science, Chengdu University of Technology, Chengdu 610000, China; zhanghuiqin@mail.cdut.edu.cn (H.Z.); chenanhang1997@163.com (A.C.)

**Keywords:** social media usage, knowledge management behavior, employee creativity, SEM

## Abstract

Employee use of social media in the workplace has become a common phenomenon. Thus, how to effectively manage and utilize employee social media usage in the workplace has become a new issue. This study examines how employees’ work-related and social-related social media usage at work can present different impacts upon their creativity through knowledge management behaviors. To test the research model, this study collected data from 425 employees in various industries in China and utilized a covariance-based structural equation model (CB-SEM) to test the hypotheses. The results suggested that work-related social media usage enhances employee creativity through promoting knowledge sharing and restraining knowledge manipulation. On the contrary, social-related social media usage cannot indirectly influence employee creativity through knowledge management behaviors. This study contributes to the literature on social media research by providing theoretical arguments on how employee use of social media for different purposes affects their creativity. Furthermore, this research offers the insight of the different paths of work-related and social-related social media usage that influence employee creativity rather than treating social media usage as a unitary concept and linking it simply with work results. This study also explores the role of three knowledge management behaviors in the relationship between social media usage and employee creativity.

## 1. Introduction

Social media has been widely used for work communication and social interaction due to its unique advantages of efficiency and convenience. Employee use of social media in the workplace has become a common phenomenon. In organizations, employees use many social media platforms for sharing and communicating, which realizes instant communication and cooperation (Yu et al., 2018) [1] and brings some benefits. Employee use of social media in the workplace contributes to the acquisition of online social capital, which can improve job performance (Huang & Liu, 2017) [2]. However, some studies have found that employees using social media to deal with personal matters at work will lead to distractions and lower productivity (Leftheriotis and Giannakos, 2014) [3], such as using social media for social and hedonic purposes (Ali et al., 2019) [4]. Thus, whether the impact of social media usage in the workplace is beneficial for employees or organizations is controversial.

At present, the research on social media usage has focused on three aspects. First, some studies have explored the impact factors of social media usage. For example, Ridings and Gefen (2004) [5] found that motivation is an important factor influencing individuals using social media. Second, some scholars have examined how social media usage influences job-related variables. Cao and Yu (2019) [6] found that the excessive use of social media leads to reduced work performance. Song et al. (2019) [7] indicated that social media usage is helpful for improving both team and individual job performance. Moreover, social media usage has been found to positively influence employee creativity via mediating variables (Wang et al., 2022) [8]. Third, a few scholars have paid attention to multiple social media platforms, such as enterprise social media (e.g., Microsoft Yammer), personal social media (e.g., Facebook, WeChat) (Song et al., 2019) [7], and public social media platforms (Van Zoonen et al., 2017) [9]. However, multiple social media platforms are frequently used for different purposes (Yu et al., 2018) [1]: employees can use them for work (Leftheriotis and Giannakos, 2014) [3]; (Van Zoonen et al., 2017) [9]; (Van Zoonen et al., 2014) [10] or social purposes (Cao et al., 2016) [11]; (Zhang et al., 2019) [12]. The various purposes of social media usage may have different influences on job-related variables. Most studies have not segmented the different purposes of social media usage. They just treat social media usage as a unitary concept. Few studies have explored the effect of social media usage from the perspective of different usage purposes in the workplace.

Allowing employees to use social media at work utilizes its superiority for information flow and transfer. Effective information dissemination helps employees gather all kinds of knowledge in a social network, which promotes innovative ideas (Alshahrani and Pennington, 2018) [13] and increases the possibility of employees producing creative work (Rhee and Choi, 2017) [14]. The ability to generate novel and useful ideas for work represents employee creativity, which is crucial for the survival and prosperity of an organization (Alshahrani and Pennington, 2018) [13]. Previous research has explored the benefits of social media usage for knowledge sharing and transfer in the workplace (Sun, et al., 2019) [15]. Knowledge sharing as a key factor in improving creativity (Eidizadeh et al., 2017 [16]; Liao & Chen, 2018 [17]) is a unified view in existing studies. However, the question of social media usage promoting knowledge flow whether indirectly enhances employee creativity or not is unclear. A lot of studies that have explored social media usage have primarily focused on knowledge sharing or knowledge exchange, often in isolation (Sun, et al., 2019) [15]. However, the use of social media, which promotes knowledge flow does not necessarily equate to a completely effective knowledge transfer. Employees may choose to share information as a matter of trust or refuse to share their knowledge because of a knowledge-sharing dilemma (Moser, 2017) [18]. For example, employees may be worried that knowledge sharing brings a threat to their own status and the shared knowledge cannot be rationally utilized by others because of other’s lack of ability (Fang, 2017) [19]. There are also knowledge-hiding behaviors. In social interactions, employees strategically choose different knowledge management behaviors to cope with a knowledge sharing request. The existing research has ignored the role of different knowledge management behaviors and just focused on a single knowledge management behavior. In order to explore the question of how social media usage enhances employee creativity, various knowledge management behaviors should be considered.

To narrow these research gaps, we investigate the relationship between different purposes of social media usage and knowledge management behaviors to determine how social media usage either facilitates or hinders employee creativity and how this works. Thus, this study focuses on exploring the following research questions:

RQ1. How does work-related and social-related social media usage affect employee creativity?

RQ2. Are there potential different impacts in three types of knowledge management behaviors for different purposes of social media usage that affect employee creativity?

## 2. Literature Review and Theoretical Background

### 2.1. Social Media Usage

As social media is an important tool for people to communicate and share, different scholars have put forward different views for its definition. Zeng and Gerritsen (2014) [20] believe that social media is an interactive tool that people use to establish contact, exchange ideas, and share information with others. Filo et al. (2015) [21] indicated that social media is a new media technology promoting interaction and cocreation by allowing the development and sharing of user-generated content. In short, social media is a web-based platform that allows individuals to share, publish, edit, categorize, and save different types of messages, information, and knowledge (Song et al., 2019) [7]. With the widespread popularity of social media, social media has penetrated deeply into the workplace (Roth et al., 2016) [22] and employee use of social media in the workplace becomes a common phenomenon.

Many scholars examined the positive impact of social media usage in the workplace, such as accumulating online social capital and improving job satisfaction (Huang & Liu 2017) [2], enhancing affective organizational commitment (Luo et al., 2018) [23], and improving team and individuals job performance (Song et al., 2019) [7]. However, other scholars have raised the concern of potential negative impacts on social media usage in the workplace. For example, Brooks (2015) [24] found that employee use of personal social media in the workplace can lead to reduced job performance, as well as increased technostress and decreased happiness (Zheng and Lee, 2016) [25]. Yu et al. (2018) [1] found that employee use of social media beyond the optimal level leads to overload, which increases the psychological pressure and reduces job performance (Cao and Yu, 2019) [6]. Moreover, using social media at work results in low productivity as employees spend too much time connecting with friends or chatting with colleagues (Wushe and Shenje, 2019) [26]. The reason why this happens is that increasing non-working connections and group discussions on social media interrupts daily workflow and increases work-life conflicts (Van Zoonen et al., 2017) [9]. In addition, employees using social media at work increases the risk of employee turnover because they have the opportunity to search for information on other companies and apply for a new job (Bizzi, 2018) [27].

There is no doubt that prior scholars have made great contributions to social media literature. However, the impact of social media usage on job-related variables is still controversy. Additionally, most scholars treated social media usage as a unitary concept. They focused on public social media usage (Van Zoonen et al., 2017) [9], personal social media usage (Brooks, 2015) [24], and enterprise social media usage (Liu & Bakici, 2019) [28] but fail to explore the use of social media by employees for different purposes. In fact, social media can be divided into social use, cognitive use, and hedonic use regarding its dimension (Ali et al., 2019) [4]. Other scholars divided social media usage into social media for work use (Leftheriotis & Giannakos, 2014 [3]; Van Zoonen et al., 2017 [9]; Van Zoonen et al., 2014 [10]), for social use (Cao et al., 2016 [11]; Zhang et al., 2019 [12]), or for work-related (Huang & Liu 2017) [2] and social-related usage (Zhang et al., 2019) [12].

Drawing from previous studies, this study focuses on work-related and social-related social media usage to explore the influence on employee creativity.

### 2.2. Knowledge Management Behavior

In a social network, any member can post a specific request for sharing knowledge or information, but other members can choose to sidestep requests (Fang, 2017) [19]. The reason why is that the shared knowledge contributes to the collective knowledge, but individuals no longer possess the proprietary value of their private knowledge (Mudambi and Navarra, 2004) [29]. In organizations, social media usage promotes fast information flow between employees. But, employees decide to contribute to the collective knowledge depends on their perceived loss and gain from sharing knowledge when they using social media for work-related communication (Moser, 2017) [18]. For example, employees may refuse to share because they worried that the shared knowledge cannot be rationally utilized by others’ lack of ability or that sharing will threaten their own status (Fang, 2017) [19]. Therefore, different knowledge management behaviors as strategical choices are made by employees in the context of trade-offs. To strategically manage one’s own knowledge, employees perform several distinct knowledge management behaviors, including knowledge sharing, knowledge hiding, and knowledge manipulation (Rhee and Choi, 2017) [14], which all reflect employees’ tactical intentions in knowledge management (Kimmerle et al., 2011 [30]; Steinel et al., 2010 [31]).

Knowledge sharing refers to the behavior of providing information and knowledge to help others (Wang and Noe, 2010) [32]^.^ Knowledge hiding stands for deliberate attempts to conceal knowledge which require to share by others (Connelly et al., 2012) [33]. Knowledge manipulation refers to deliberately exaggerating in the value and content of knowledge for one’s own benefit (Rhee and Choi, 2017) [14]. Employees share knowledge by exchanging task-related ideas, information, and knowledge that their colleagues need (Wang and Noe, 2010) [32]. In this way, they exchange knowledge and collaborate with colleagues to create new knowledge (Eidizadeh et al., 2017) [16]. It is worth noting that knowledge sharing and knowledge hiding are not opposite behaviors. Knowledge hiding is a deliberate concealment or suppression of easily acquired knowledge, while low-level knowledge sharing is mainly a sharing behavior caused by lack of knowledge (Connelly et al., 2012) [33]. They are fundamentally different. Knowledge hiding includes evasive hiding, playing dumb, and rationalized hiding (Connelly et al., 2012) [33]. While evasive hiding refers to a behavior or a misleading promise that knowledge providers are providing other information rather than what is really requested, playing dumb reflects knowledge providers pretending not to know the relevant knowledge, and rationalized hiding describes knowledge providers indicating that they are unable to provide the requested knowledge due to specific reasons (Connelly et al., 2012) [33]. Existing researches attempted to explore how employees manage their knowledge in social interactions, but mainly focus on a single knowledge management behavior, such as knowledge sharing (Kwahk and Park, 2016) [34]. A few studies attend to multiple types of knowledge management behaviors. This study will focus on exploring the role of three different knowledge management behaviors in the relationship between social media usage and employee creativity.

### 2.3. Employee Creativity

Employee creativity is creative methods and ideas that employees used to solve work-related problems (George and Zhou, 2001) [35], which comes from knowledge holding and exchange among colleagues (Rhee and Choi, 2017) [14]. In other words, employee creativity is identified as employee use of certain cognitive processes in creative problem solving (Hughes et al., 2018) [36]. The existing research indicated that to explore in-depth understanding of how cognitive process facilitates creativity, the important factors, which developed at multilevel perspective (e.g., individual, team, and group level) (Reiter-Palmon et al., 2015) [37], need to be considered. In fact, a large number of studies have been conducted on individual factors, such as self-efficacy (Magadley and Birdi, 2012) [38], and organization-level factors, such as organizational culture (Ogbeibu et al., 2018) [39] and leadership style, (Hughes et al., 2018) [36] to examine the impact of various factors on creativity. For example, Zhou et al. (2018) [40] explored the influence of visionary leadership on employee creativity. However, individual creativity is also affected by the social network it is embedded in, which is at the meso level (Sigala and Chalkiti, 2015) [41]. Nowadays, the popularity of social media has greatly changed the way people share knowledge, communicate, and cooperate (Filo et al., 2015) [21]. Reiter-Palmon et al. (2015) [37] indicated the core cognitive process relevant to creativity, including the idea-generation phase (problem identification and construction, information search and encoding, and idea and solution generation) and the implementation phase (idea evaluation and selection, and implementation planning and monitoring). Social media usage enables individuals to engage in a continuous exchange of ideas in an informal manner with others with the same interests. They share and integrate knowledge from various sources to create new metaknowledge and participate in the collective knowledge-generation process (Sigala and Chalkiti, 2015) [41]. It promotes the development of the core cognitive process relevant to creativity, including information search and encoding, idea and solution generation, and idea evaluation and selection. Moreover, Mumford and Gustafson (1988) [42] argued that the major and minor creative contributions both require several different knowledges, skills, and abilities. Social media usage provides the opportunity for individuals to search for and gather different knowledge. Meanwhile, there are quite a lot of studies that have verified that social media usage has an influence on employee creativity. For instance, Korzynski et al. (2019) [43] showed that social media usage can promote employee creativity through online social knowledge management. Enterprise social media usage also improves employee creativity with the mediating effect of leader–member exchange and support for innovation (Wang et al., 2022) [8]. Nevertheless, the current studies neglect how the different purposes of social media usage influences employee creativity from a knowledge management perspective. This study will answer this question by exploring the relationship of social media usage, employee creativity and knowledge management behavior and provide insights for future research.

### 2.4. Theoretical Foundation

#### 2.4.1. Uses and Gratification Theory

To specify the effect of social media usage on employee creativity, we focus on the different purposes of social media usage, which includes work-related and social-related social media usage. Employees’ choice to use social media for work or social purpose is driven by their desire to satisfy their wide range of needs. According to the explanation of uses and gratification theory, it is beneficial to understand the relationship between the different purposes of social media usage and related work outcomes. The uses and gratification theory holds that people use media in different purposes and that they are active in selecting the information resources they are willing to access (Liang et al., 2006) [44]. It is mostly used to illustrate how users use media to meet their personal needs for different purposes (Li et al., 2018) [45]. This theory has been widely applied to various media, including internet media, blogs, online games (Li et al., 2015) [46], virtual communities (Liu et al., 2017) [47], and social networks (Johnson and Kaye, 2015) [48]. For example, Huang and Zhou (2018) [49] utilized the uses and gratification theory to explain users’ behavior on mobile shopping media platforms. Ali et al. (2019) [4] discussed the impact of social media usage on team innovative performance based on the uses and gratification theory. In brief, the uses and gratification theory provide a link between choice and outcome, illustrating that users’ choice of certain media depends on the satisfaction of needs (Stafford et al., 2004) [50]. Therefore, this paper makes use of the uses and gratification theory to explore how employees using social media for different purposes to meet their own needs will affect their work results. In other words, this study examines the association of social media usage (work-related and social-related social media usage), employee behavior, and outcomes based on the uses and gratification theory.

#### 2.4.2. Connectivism Learning Theory

We also investigate the distinct effects of the three knowledge management behaviors on employee creativity in the context of social media usage. Connectivism learning theory is a new learning theory in the digital area, which is related to the application of information and communication technology (Sitti et al., 2013) [51]. It was proposed by Siemens (2005) [52] to describe the linkage of human learning and the ubiquitous knowledge acquisition that take place within the underlying technological environment. According to the point of connectivism learning theory, it is applicable to explain the relationship between knowledge management behavior and social media usage. Connectivism learning theory interprets the important role of networking technologies for learning and the process of knowledge creation and sharing in an online context (Frederique and Elio, 2020) [53]. Moreover, it highlights that what is vital for learning is a social network that people have access to rather than what they know (Siemens, 2005) [52]. Connectivism learning theory also emphasizes that knowledge creation is based on the combination of internal and external cognitive processes in social networks (Sigala and Chalkiti, 2015) [41]. Social media is a basic tool of social interaction, and its usage reflects the extension of a personal network, which amplifies learning and knowledge creation. Coincidentally, social interaction and communication realize the connection between internal and external cognitive processes within social networks (Sigala and Chalkiti, 2015) [41]. Social media usage empowers people to process a massive amount of information, which expands their learning processes and knowledge acquisition in the social network. Utilizing connectivism learning theory to support how social media usage influence knowledge management is suitable.

## 3. Research Models and Hypotheses

### 3.1. Social Media Usage and Knowledge Management Behavior

#### 3.1.1. Social Media Usage and Knowledge Sharing

Knowledge sharing is a cognitive and behavioral process, including individuals’ consciousness construction in specific knowledge fields and the understanding of knowledge based on own experience and the environment they live in (Yeo and Marquardt, 2015) [54]. In organizations, knowledge sharing is regarded as an important process of social interaction (Razmerita et al., 2016 [55]; Yeo and Marquardt, 2015 [54]). It provides opportunities for knowledge exchange. At work, employees exchange knowledge through contribution, collection, offering help, and replying to messages (Chen et al., 2020) [56]. For example, employees actively communicating and consulting with colleagues can be regarded as the process of contributing knowledge and collecting knowledge (Razmerita et al., 2016) [55]. Knowledge sharing involves people consciously providing their knowledge to others (Ipe, 2003) [57], which is influenced by extrinsic and intrinsic motivations (Gagné et al., 2019) [58]. Employees exchanging task-related ideas and information with colleagues and providing knowledge that colleagues need can realize knowledge sharing at work (Wang and Noe, 2010) [32].

The use of social media at work improves the intensity, frequency, and breadth of knowledge exchange among employees. Guided by a common vision, worked-related social media usage promotes communication with colleagues and the integration of different resources (Cao et al., 2016) [11], which is conducive to the completion of work tasks. Social media usage also helps employees to establish and maintain relationships with others through social activities, thus achieving knowledge exchange (Luo et al., 2018) [23]. Social-related social media usage enables employees to maintain social relations embedded in social networks by contacting family, friends, and acquaintances, thus obtaining social support and a sense of belonging (Cao et al., 2016) [11]. Whether to achieve work goals or obtain social support and a sense of belonging, work-related and social-related social media usage both enables employees to adopt knowledge sharing as a coping strategy when faced with the demands of social interaction. Therefore, based on the above arguments, we propose the following hypotheses:

**H1:** 
*Work-related social media usage significantly and positively influences knowledge sharing.*


**H2:** 
*Social-related social media usage significantly and positively influences knowledge sharing.*


#### 3.1.2. Social Media Usage and Knowledge Hiding

According to the definition of Connelly et al. (2012) [33] on knowledge hiding, knowledge hiding in this study refers to the behavior of employees who intentionally do not reply to or inform others of the real information that they have requested. Knowledge hiding includes evasive hiding, playing dumb, and rationalized hiding (Connelly et al., 2012) [33]. In relation to social media, evasive hiding can be achieved by sharing irrelevant information rather than valid knowledge; playing dumb can be achieved by pretending to be busy or offline by turning off chat rooms, setting group messages to do not disturb, and not reading unwanted messages; and rationalized hiding is possible by controlling access to shared documents (Fang, 2017) [19]. The main reasons trigger knowledge hiding coming from individual factors, organizational factors, and situational factors. Studies focus on individual factors showed that individual traits (e.g., Machiavellianism, narcissism) (Pan et al., 2018) [59], goal orientation (Rhee and Choi, 2017) [14], lack of self-confidence (Jha and Varkkey, 2018) [60] can predict knowledge hiding behavior. Organizational factors, such as perceived occupational insecurity Jha and Varkkey, 2018) [60] or job insecurity (Serenko and Bontis, 2016) [61], as well as interpersonal distrust in situational factors (Jha and Varkkey, 2018) [60] have a positive impact on knowledge hiding. Employees may adopt knowledge hiding as a coping strategy for any of the above reasons when facing knowledge sharing requests. However, compared with traditional face-to-face communication, social media is an effective tool for information exchange, it provides more convenient functions and observable clues, such as real-time interaction, user status, data access, and storage (Fang, 2017) [19]. Sheer and Rice (2017) [62] pointed out that employees heavily rely on mobile instant messaging software to communicate with colleagues and clients. Work-related social media usage ensures employees can communicate with colleagues on social media and complete the exchange of work-related information (Sun and Shang, 2014) [63]. Meanwhile, because of the achievement of work goals, employees also pay attention to the dynamics of the work group to understand the work tasks in real-time. Social-related social media usage requires employees to browse their messages in real-time to maintain social relations embedded in social networks (Cao et al., 2016) [11]. In addition, the popularity of social media provides a platform for employees to share, and they are willing to share what they saw and heard through social media (such as sharing moments). Based on the above views, we propose the following hypotheses:

**H3:** 
*Work-related social media usage significantly and negatively influences knowledge hiding.*


**H4:** 
*Social-related social media usage significantly and negatively influences knowledge hiding.*


#### 3.1.3. Social Media Usage and Knowledge Manipulation

As a new topic in knowledge management research, knowledge manipulation has not been clearly defined. Existing studies pointed out that knowledge manipulation means deliberately exaggerating the value and content of knowledge for own benefit (Rhee and Choi, 2017) [14]. Bettis-outland (1999) [64] indicated that individuals do not always actively share information and knowledge when they transmit information and knowledge, and sometimes the information transmission and knowledge exchange will be artificially distorted to seek their interests. According to the above definition, we believe that knowledge manipulation is a flexible coping strategy adopted by individuals. In other words, when faced with knowledge sharing requests, individuals will share their knowledge in misleading ways to maximize their interests.

In this study, knowledge manipulation is regarded as a kind of avoidance response, which is another coping strategy besides knowledge sharing and knowledge hiding. Similar to knowledge sharing and knowledge hiding, knowledge manipulation may also occur when using social media. When colleagues ask someone for work-related content, someone may ignore potential problems in the shared knowledge (Rhee and Choi, 2017) [14]. Ford and Staples (2010) [65] indicated that partial knowledge sharing occurs more often than full knowledge sharing in real situations and that partial knowledge sharing only involves sharing some relevant knowledge while ignoring content that comes with risks. Akin to knowledge hiding, work-related social media usage is associated with employees achieving work goals. Moreover, compared with traditional communication methods, it is easier to verify whether information is distorted. So, we believe that work-related social media usage is not conducive to employees adopting knowledge manipulation. Social media is designed to satisfy the entertainment of individuals, and users will voluntarily share real content. Therefore, we propose the following hypotheses:

**H5:** *Work-related social media usage significantly and negatively influences knowledge manipulation*.

**H6:** 
*Social-related social media usage significantly and negatively influences knowledge manipulation.*


### 3.2. Knowledge Management Behavior and Employee Creativity

#### 3.2.1. Knowledge Sharing and Employee Creativity

Knowledge sharing refers to the behavior of sharing information and ideas with others (Elrehail et al., 2016) [66] and gathering knowledge to create new ideas through knowledge exchange (Eidizadeh et al., 2017) [66]. Wang and Noe (2010) [32] summarized knowledge sharing as a process of actively expressing one’s own ideas, sharing one’s own information to deal with problems for others, or discussing innovative methods and developing new ideas through cooperation. The widespread use of social media makes knowledge widely spread in virtual network communities and promotes knowledge sharing activities in social networks (Kwahk and Park, 2016) [34]. Those knowledge sharing activities stimulate creativity by triggering divergent thinking (Rhee and Choi, 2017) [14]. Therefore, the use of social media promotes the flow and acquisition of knowledge from social networks, which in turn promotes individual creativity through communication. In organizations, effective knowledge sharing among organization members can reduce the cost of knowledge production and ensure the sharing of best practices, so that organizations can solve practical problems (Eidizadeh et al., 2017) [66]. Knowledge sharing activities transform tacit knowledge into explicit knowledge (Rhee and Choi, 2017) [14], which helps employees adopt creative approaches to problem solving and propose best practices from others’ experiences (Edwards et al., 2017) [67]. Many organizations encourage employees to share knowledge online because it promotes the effective flow and wide dispersion of knowledge among employees (Pee and Lee, 2015) [68]. Social media usage creates a virtual space that supports knowledge sharing activities, promotes knowledge sharing in social networks, and ensures the wide spread of knowledge among individuals, communities, and society (Kwahk and Park, 2016) [34]. Therefore, social media usage enables employees to exchange their thoughts, ideas, viewpoints, and emotions with others and learn from each other through communication and interaction, which is conducive to improving their knowledge reserve and innovation potential. When using social media for work or social interaction, employees can access information and knowledge shared by others to help them deal with problems. Therefore, we propose the following hypothesis:

**H7:** 
*Knowledge sharing significantly and positively influences employee creativity.*


#### 3.2.2. Knowledge Hiding and Employee Creativity

Combined with the definition of knowledge hiding, we believe knowledge hiding is not beneficial to the improvement of employee creativity. First, knowledge hiding separates employees from social networks where they interact (Connelly et al., 2012) [33]. Creativity highly relies on the exchange and sharing of information (Černe et al., 2014) [69]. The act of being excluded makes employees access to limited knowledge. It means that employees only focus on their own perspectives and knowledge; they cannot enter the collective knowledge network. Their personal ability to produce creative results is limited, and the lack of social interaction with others will cause them unable to come up with innovative ways to solve urgent problems at work (Rhee and Choi, 2017) [14]. Secondly, according to the social exchange theory, when employees hide knowledge, it triggers a cycle of mutual distrust. As a response to the perceived negative response, coworkers are also unwilling to share knowledge with them (Černe et al., 2014) [69]. Existing studies showed that knowledge hiding has adversely affected employee creativity. For example, Rhee and Choi (2017) [14] showed that knowledge hiding has a negative relationship with employee creativity. Malik et al. (2019) [70] found that employees perceived organizational politics can affect employee creativity by influencing knowledge hiding. We believe that employees will also face situations of knowledge hiding that affect their creativity in the process of using social media. Based on previous studies, we propose the following hypothesis:

**H8:** 
*Knowledge hiding significantly and negatively influences employee creativity.*


#### 3.2.3. Knowledge Manipulation and Employee Creativity

At present, there are few studies on knowledge manipulation. Rhee and Choi (2017) [14] pointed out that knowledge manipulation as a knowledge management strategy has a positive impact on employee creativity in their study of exploring the different knowledge management behavior strategies adopted by employees due to their different goal orientations. According to the definition of knowledge manipulation mentioned above, we believe that the knowledge shared by sharers through knowledge manipulation is not real information and that it deviates from accurate and true information, which means that the knowledge provided by the knowledge manipulators is misleading. So, we speculate that knowledge manipulators will also encounter negative reciprocity when sharing knowledge. This is because when knowledge recipients perceive that the other parties exchange knowledge with a disingenuous attitude, they may respond in the same way, which leads to uncertainty in information exchange and knowledge sharing. In other words, knowledge manipulators cannot guarantee the accuracy of information obtained from others. Employees exchanging ineffective information with others is unhelpful in generating new ideas. Based on the above views, we propose the following hypothesis:

**H9:** 
*Knowledge manipulation significantly and negatively influences employee creativity.*


Drawing upon the above arguments, Figure 1 shows the research model. It displays the paths of how work-related and social-related social media usage influence employee creativity through different knowledge management behaviors.

## 4. Methods

### 4.1. Procedure and Participants

#### 4.1.1. Procedure

To test the proposed theoretical model and the hypotheses, an online questionnaire was conducted to collect data. We carried out our research in China and follow up with a back translation to ensure semantic consistency. Three researchers from different backgrounds (with English as a native language or official language) were invited to read the original English scale and the translated Chinese scale. According to their opinions, rational adjustment and modification of the items are conducted. Then, we sent the final translated scale to 5 employees with experience in using social media to review the logical consistency and contextual relevance. Suitable modifications were made based on their feedback. Considering that this study examines the potential impact of social media usage on employee creativity in the workplace, we target the respondents as employees with social media using experience in the workplace. In practice, employees use multiple social media software at work. To ensure all participants had experience with social media usage, we set a screening question in the survey “Have you ever used social media at work? (Such as QQ, WeChat, TIM, Weibo, Zhihu, and others)”. Only those who chose “Yes” could fill in the questionnaire.

Finally, we employ a professional market research company to collect data. This company is a leader in market research in China and has more than 2,600,000 members to participate in various research projects via strict recruiting methods. To ensure all participants are our target people, we put forward special requirements requiring enterprise employees as the target participants. To control the quality, each respondent can only submit one complete questionnaire, and IP addresses are limited to one submission. In the end, we collected a total of 550 complete responses, and 105 invalid responses were eliminated systematically. In addition, 20 invalid responses were removed by manual review. Ultimately, a total of 425 valid questionnaires were collected. Table 1 presents the descriptive statistics of all respondents.

#### 4.1.2. Participants

During the survey, all participants were informed in written that their participation is voluntary. In the introduction section of the questionnaire, we briefly explained the background and purpose of the survey, emphasizing that participants can voluntarily choose to participate in the questionnaire survey or not. The introduction states “This questionnaire aims to understand the impact of social media usage in daily work, if you are willing to participate in the survey, please select ‘YES’ and complete the survey, if not, please select ‘NO’ and withdraw from the survey.” When collecting data, the questionnaire data of some participants who choose not to participate in the survey and fail to complete all the questions were eliminated systematically. Data were collected only from participants who agreed to participate in the survey and completed all questions in the questionnaire. Furthermore, the questionnaire data of participants were removed by manual review based on respondents taking less than 2 minutes to fill in the questionnaire and choosing the same option for almost all the options.

The employee samples included 53.2 percent of male participants. For ease of collecting the age of participants, we defined “1 = under 25 years old, 2 = 26–30 years old, 3 = 31–35 years old, 4 = over 35 years old”, for which the average is 2.21 (SD = 0.773). The following are the ages of the participants: under 25 years old (16.5 percent), 26–30 years old (50.6 percent), 31–35 years old (28.0 percent), and over 35 years old (4.9 percent). The following are educational levels of the participants: associate degree or below (16.7 percent), bachelor’s (72.5 percent), and master’s/PhD (10.8 percent).

### 4.2. Measures

The tested scales used in this study were adapted from existing validated measures in previous studies. Work-related social media usage and social-related social media usage were measured using a 9-item scale developed by Gonzalez (2012) [71]. Among them, 5 items were used to measure work-related social media usage, and 4 items were used to measure social-related social media usage. This measurement scale of social media usage is also used in the research that Zhang et al. (2019) [12] conducted. Employee creativity was measured by the scale developed by Zhou and George (2001) [72], which contains 13 items. As for knowledge management behaviors, this study focuses on knowledge sharing, knowledge hiding, and knowledge manipulation. Knowledge sharing was measured using a 5-item scale developed by Lu et al. (2006) [73] and Bock and Kim (2002) [74], which was used to evaluate the degree that people are willing to share their knowledge. Knowledge hiding was measured using a 12-item scale developed by Connelly et al. (2012) [33], which contains three dimensions, including evasive hiding, playing dumb, and rationalized hiding. Knowledge manipulation was measured using a 4-item scale constructed by Rhee and Choi (2017) [14]. All measurement scale items are shown in Appendix A.

Previous studies found that demographic variables (Shin and Zhou, 2007) [75], such as age, gender, and education level have an influence on employee creativity. Therefore, this study includes these demographic variables as control variables in the research and the subsequent statistics.

### 4.3. Statistical Analysis

The covariance-based structural equation model (CB-SEM), as opposed to the partial least square structural equation model (PLS-SEM) was used to verify research models and hypotheses. While comparing with PLS-SEM focuses on causal prediction, CB-SEM is applicable to causal path analysis. This study focuses on causal testing, which satisfied the CB-SEM usage criteria. Therefore, CB-SEM was used, and Mplus 7.4 software was adopted. 

There was potential common method bias that exists in self-reported data (Podsakoff et al., 2003) [76]. An exploratory principal axis factoring analysis without rotation was applied to all multi-item measures (Malhotra et al., 2006) [77] in IBM SPSS 21.0. The result showed that one factor explained only 28% of the variance. Moreover, the correlation matrix (Table 2) indicated that no correlation between the variables exceeded the threshold of 0.90 (Bagozzi et al., 1991) [78]. Therefore, no common method bias occurred in the data of this study. Collinearity diagnostics was also conducted to detect multicollinearity in the research model. The variance inflation factors (VIF) for each construct was between 1.4–1.8, less than the threshold of 5 (Hair et al., 2011) [79]. This indicated no collinearity occurred among any constructs.

## 5. Result

### 5.1. Measurement Model

The measurement model can be assessed by examining instrument validity and reliability. According to the criteria suggested by Fornell and Larcker (1981) [80], construct reliability was determined by Cronbach’s α and composite reliability (CR). Cronbach’s α and composite reliability (CR) are acceptable when they exceed 0.7. Convergent validity was evaluated by the item loadings. The item loading is not less than 0.6 (Fornell and Larcker, 1981 [80]; Van Dyne, et al., 2002 [81]). Table 3 shows that the Cronbach’s α and CR for all constructs are above 0.70 and that the Cronbach Alpha for the total scale is 0.813. All item loadings ranged from 0.61 to 0.83. Thus, construct reliability and convergent validity of our measurement instrument were acceptable. Meanwhile, this paper examined the discriminant validity by checking if the square root of AVE for each construct is larger than the correlations between the construct and all other constructs (Fornell and Larcker, 1981) [80]. Table 4 indicates that the square root of each factor’s AVE exceeds the correlations with other constructs. It demonstrated confirmation of sufficient discriminant validity for study measures. Construct validity was assessed by a fit index through confirmatory factor analysis. Table 5 shows the results of the confirmatory factor analysis. The chi-square/degree of freedom ratio was 2.32, which is <5. The comparative fit index (CFI) and Tucker–Lewis index (TLI) scores were near 0.9, and the standardized root mean square residual (SRMR) was <0.08. The root means square error of approximation (RMSEA) was <0.08. These results met the criteria of accepted model of fit, indicating that the confirmatory factor analysis (CFA) model fit well.

### 5.2. Structural Model

The structural model results are illustrated in Figure 2. This study adopts bootstrap resampling estimation to test hypotheses because it has higher statistical validity (Cheung and Lau, 2008 [82]; Williams and MacKinnon, 2008 [83]), and it is the best method for testing mediating effects (Preacher and Hayes, 2008) [84]. Scholars suggested that model tests should consider research purpose, theoretical background (Barrett, 2007 [85]; Bollen, 2011 [86]), and the significance of parameter estimates (Cole and Maxwell, 2003) [87], such as the factor loadings for effect indicators and the regression coefficients for causal indicator (Bollen, 2011) [86], and the confidence intervals around key parameter estimates (Cole and Maxwell, 2003) [87]. Therefore, the structural model results were assessed via the estimated path coefficients with asterisks. The results showed that H1, H3, H5, H7, and H9 were supported, but H2, H4, H6, and H8 were not. Specifically, work-related social media usage significantly and positively affects knowledge sharing (β = 0.368, 95% [0.080, 0.773]); work-related social media usage significantly and negatively affects knowledge hiding (β = −0.543, 95% [−1.610, −0.077]) and knowledge manipulation (β = −0.596, 95% [−1.728, −0.053]). However, the effect of social-related social media usage on knowledge sharing (β = 0.081, 95% [−0.396, 0.349]), knowledge hiding (β = 0.138, 95% [−0.343, 1.156]), and knowledge manipulation (β = 0.376, 95% [−0.173, 1.521]) was not significant. Thus, H2, H4, and H6 were all rejected. Moreover, knowledge sharing was significantly and positively correlated with employee creativity (β = 0.547, 95% [0.480, 0.612]), but the impact of knowledge hiding on employee creativity was not significant (β = 0.043, 95% [−0.024, 0.115]). Knowledge manipulation significantly and negatively affects employee creativity (β = −0.104, 95% [−0.160, −0.053]). Figure 2 presents the structural model with standardized path coefficients. Table 6 provides the bootstrapping estimates for path analysis, including the confidence intervals.

## 6. Discussion and Implications

### 6.1. Discussion

This study examines the relationship between social media usage, knowledge management behaviors, and employee creativity, and puts forward nine related hypotheses. After hypothesis testing, part of the hypotheses is verified.

The results show that work-related social media usage has a significant positive impact on knowledge sharing as well as a significant negative impact on knowledge hiding and knowledge manipulation. It means work-related social media usage promotes knowledge sharing among employees and reduces knowledge hiding and knowledge manipulation. This result is consistent with previous studies, which indicated that employees using social media for work purposes promotes communication and knowledge sharing with colleagues on the basis that they share a common vision and goal (Cao et al., 2016 [11]; Luo et al., 2018 [23]). At the same time, because work-related social media usage is mainly used for better and more efficient completion of work tasks, knowledge hiding and knowledge manipulation behaviors are relatively reduced. This is also in accordance with the explanation of social exchange theory that the social process involves an exchange between individuals (Singh et al., 2018) [88] through interaction and communication (Wang et al., 2022) [8]. Social-related social media usage has no significant impact on knowledge sharing, knowledge hiding, and knowledge manipulation behaviors. We attribute this result to the social function of social media. The social function of social media is to build and maintain social relationships (Ali et al., 2019) [4]. It means social-related social media usage is mainly for personal purposes, such as maintaining social relations and enhancing connections and emotions with others. Employee use social media for social purposes, such as to chat with colleagues or friends, is mainly about emotional communication, and the exchange of work-related knowledge may be less than what is expected.

In addition, this study also explores the impact of three types of knowledge management behaviors on employee creativity. The results show that knowledge sharing has a significant positive impact on employee creativity, knowledge hiding has no significant impact on employee creativity, and knowledge manipulation has a significant negative impact on employee creativity. Knowledge sharing is a key factor affecting creativity, as it can significantly predict employee creativity (Marianna and Kalotina, 2015 [41]; Eidizadeh et al., 2017 [16]; Liao and Chen, 2018 [17]), which is a unified view in existing studies. The results of this study are consistent with most previous studies. However, previous research has verified that knowledge hiding has a negative impact on employee creativity, while knowledge manipulation has a positive impact on employee creativity (Rhee and Choi, 2017) [14]. This study shows there is no significant effect between knowledge hiding and employee creativity. On the one hand, we think that this may be due to the different context. This study investigates whether employees adopt knowledge hiding to reply to others’ knowledge sharing requests in a social-media-usage context. It is different from previous research, which have focused on the context of in face-to-face communication. On the other hand, we believe that employees who use social media may always adopt rationalized hiding as the main way of knowledge hiding, which leads to different results. As the third knowledge management strategy, which lies between knowledge sharing and knowledge hiding, knowledge manipulation may lead to uncertainty and inaccuracy in information exchange and knowledge sharing. Therefore, there may be misunderstandings during knowledge exchange between the knowledge providers and knowledge receivers, which will have a reasonable negative impact on employee creativity.

### 6.2. Theoretical Implications

The theoretical contribution of this study is reflected in two aspects. It expands the research on the antecedent variables of employee creativity and the application scope of existing relevant theories in social media research. First, existing studies mostly focused on enterprise social media usage or simply treated social media usage as a unitary concept, which makes it difficult to clearly understand the specific effects of social media usage in the workplace. Second, there is little research exploring the effect of social media usage on employee creativity. From the perspectives of different purposes of social media usage, this study discussed the influence of work-related and social-related social media usage at work on employee creativity, providing a deep understanding of how social media usage influences employees’ work results, which extends the research on antecedent variables of employee creativity. Furthermore, this study utilizes the uses and gratification theory and connectivism learning theory to explore the impact of social media usage on employee creativity and contextualizes relevant theories to enrich and expand research on relevant theories. Finally, this study introduces theories in the field of psychology into management research. To a certain extent, it expands the application scope of the theory and widens the theoretical perspective about the research on the employee creativity mechanism. Moreover, it deepens awareness and understanding of employees’ use of social media in the workplace.

### 6.3. Practical Implications 

This study provides practical suggestions for managers, indicating that social media usage for different purposes in the workplace will have different impacts on employee creativity. Managers cannot simply view social media usage at work as a bad usage pattern. Managers should properly understand employees use of social media in the workplace. Therefore, both employees and organizations should pay attention to the impact of social media usage at work. This study mainly provides the following enlightenment.

Firstly, employees should be aware of the positive results of work-related social media usage. They should realize that they can gain knowledge by communicating with others. For example, employees can ask colleagues or peers questions in their social networks and ask them to share their knowledge. In addition, employees should be aware of using social media to obtain shared documents, data, and other resources from colleagues, which helps improve their work efficiency and enrich their knowledge reserve. Employees also can use social media to search, collect work-related information, and edit content to share for gaining feedback. By managing a large amount of fragmented knowledge, the knowledge can be used by themselves.

Secondly, this study shows that work-related social media usage can promote employee creativity by facilitating their knowledge sharing behavior. Organizations should be aware of that social media provides a platform for employees to access information and knowledge (Alshahrani and Pennington, 2018 [44]; Panahi et al., 2016 [89]). Organizations can encourage employees to communicate on social media, rather than simply restrict social media use in the workplace. Additionally, organizations should also realize that employees use social media for work- and social-related matters that cannot simply be separated in the current work environment. Formulating policies to stress work-related social media usage rather than social-related social media usage at work is needed. In addition, the results of this study indicate that work-related social media usage can inhibit employees’ knowledge manipulation behavior and promote knowledge sharing behavior. Organizations can support employees using social media to share work-related information by establishing online groups based on social media to create a virtual community of practice (Bandow and Gerweck, 2015) [90] for collective knowledge contribution. The results of this study also show that the influence of work-related social media usage on knowledge manipulation is greater than knowledge sharing. Thus, encouraging and guiding employees to correctly use social media to interact with colleagues is conducive to the effective dissemination of knowledge.

### 6.4. Limitations and Future Research

Although this study has made useful supplements and some contributions to existing relevant studies, it still has limitations, and we hope that further exploration can be carried out in future studies.

Firstly, the research needs to be further deepened. This study divides social media usage into work-related and social-related social media usage according to different purposes of social media use. But this classification covers all non-work-related social media use patterns in social-related usage. Future studies can explore other dimensions of social media usage, such as cognitive use, hedonic use, and social use (Ali-Hassan et al., 2015) [91]; clearly distinguish the different purposes of social media usage; and enhance the understanding of employee social media usage in the workplace. Secondly, this study explores how different knowledge management behaviors influence the relationship between social media usage and employee creativity. However, individual differences in social media usage may be related to the “Big Five” personality traits (Montag et al., 2018) [92], and personality traits may also be related to the knowledge management behaviors adopted by employees. For example, narcissism may influence knowledge hiding behaviors (Pan et al., 2018) [59]. Future studies can focus on the impact of individual differences on social media users’ behavior. Thirdly, the data in this study were collected from Chinese employees. China’s unique culture of “Guanxi” culture (Chen et al., 2017) [93] may have an additional influence on employees’ willingness to share information in social interaction. Future research should consider the influence of traditional culture. Finally, this study treats knowledge hiding as a unitary concept and finds that knowledge hiding has no impact on employee creativity. Future research can explore the different influence of three dimensions (evasive hiding, playing dumb, and rationalized hiding) on employee creativity in social media usage.

## 7. Conclusions

This study examines the association of social media usage (work-related and social-related social media usage), knowledge management behavior, and employee creativity in enterprise staff with social media usage experience at work. It investigates the different paths by which social media usage influences employee creativity through knowledge sharing, knowledge hiding, and knowledge manipulation. To empirically assess the research model, survey data were gathered from 425 employees. Some of the proposed hypotheses were supported. The results demonstrate that work-related social media usage can promote employee creativity through influence knowledge sharing and knowledge manipulation. However, social-related social media usage cannot indirectly influence employee creativity through knowledge management behavior. These research findings help enrich social media research and improve the current understanding of how social media usage influences employees’ work results and the implications of its influence.

## Figures and Tables

**Figure 1 behavsci-13-00601-f001:**
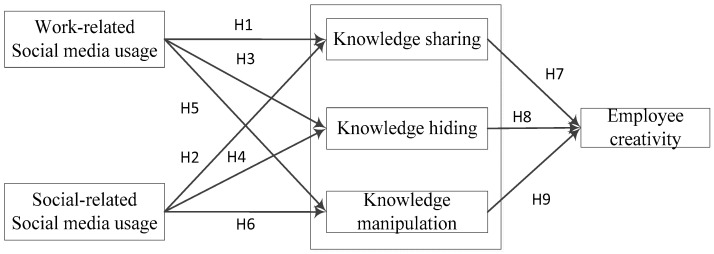
Research model.

**Figure 2 behavsci-13-00601-f002:**
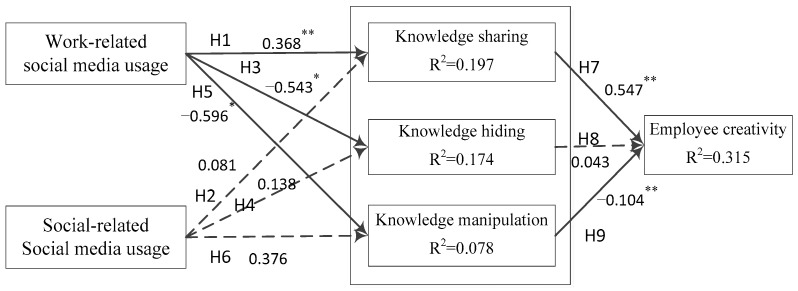
Results of the structural model. Notes: * *p* < 0.05; ** *p* < 0.01.

**Table 1 behavsci-13-00601-t001:** Respondents’ demographics.

Demographics	Items	Percent (%)
Gender	Male	53.2
	Female	46.8
Age	<25	16.5
	26–30	50.6
	31–35	28.0
	>35	4.9
Education level	Associate degree or below	16.7
	Bachelor’s	72.5
	Master’s/PhD	10.8

**Table 2 behavsci-13-00601-t002:** Correlation matrix among study variables.

	Gender	Age	Education Level	WRSM	SRSM	KS	KH	KM
Gender								
Age	−0.132 **							
Education level	0.069	0.082						
WRSM	−0.061	0.030	0.087					
SRSM	−0.084	0.020	0.044	0.635 **				
KS	−0.117 *	0.082	0.053	0.426 **	0.387 **			
KH	−0.082	−0.094	−0.129 **	−0.292 **	−0.228 **	−0.425 **		
KM	−0.075	−0.087	−0.034	−0.148 **	−0.077	−0.334 **	0.615 **	
Creativity	−0.149 **	0.125 **	0.174 **	0.390 **	0.349 **	0.654 **	−0.288 **	−0.316 **

Notes: * *p* < 0.05; ** *p* < 0.01.

**Table 3 behavsci-13-00601-t003:** Mean, SD, reliability, and convergent validity.

Construct	Item	Mean	SD	Standard Loading	Cronbach α	CR	Cronbach α (Total Scale)
Work-related social media (WRSM)	WRSM1	4.29	0.76	0.67	0.769	0.773	0.813
	WRSM2	4.13	0.68	0.62			
	WRSM3	4.20	0.76	0.61			
	WRSM4	4.09	0.72	0.61			
	WRSM5	4.23	0.65	0.67			
Social-related social media (SRSM)	SRSM1	4.16	0.74	0.61	0.737	0.741	
	SRSM2	4.29	0.68	0.69			
	SRSM3	4.21	0.71	0.68			
	SRSM4	4.14	0.80	0.60			
Knowledge sharing (KS)	KS1	4.07	0.69	0.69	0.858	0.860	
	KS2	4.17	0.66	0.70			
	KS3	4.14	0.72	0.77			
	KS4	4.04	0.72	0.83			
	KS5	4.04	0.77	0.72			
Knowledge hiding (KH)	KH1	2.18	0.93	0.78	0.931	0.932	
	KH2	2.08	0.99	0.76			
	KH3	1.96	0.92	0.78			
	KH4	2.18	1.03	0.65			
	KH5	1.87	0.88	0.77			
	KH6	1.98	0.93	0.81			
	KH7	1.87	0.87	0.78			
	KH8	2.35	1.08	0.71			
	KH9	2.44	1.12	0.68			
	KH10	1.94	0.94	0.74			
	KH11	2.46	1.06	0.67			
	KH12	2.85	1.25	0.60			
Knowledge manipulation (KM)	KM1	2.40	1.02	0.66	0.766	0.769	
	KM2	2.52	1.14	0.79			
	KM3	3.05	1.16	0.61			
	KM4	2.65	1.12	0.63			
Creativity	Creativity 1	4.04	0.64	0.73	0.912	0.913	
	Creativity 2	4.21	0.68	0.65			
	Creativity 3	4.13	0.71	0.63			
	Creativity 4	4.18	0.68	0.64			
	Creativity 5	4.02	0.76	0.73			
	Creativity 6	3.69	1.00	0.53			
	Creativity 7	4.09	0.71	0.63			
	Creativity 8	4.20	0.62	0.62			
	Creativity 9	4.20	0.72	0.61			
	Creativity 10	3.97	0.80	0.70			
	Creativity 11	4.10	0.68	0.73			
	Creativity 12	4.19	0.64	0.73			
	Creativity 13	4.03	0.75	0.74			

**Table 4 behavsci-13-00601-t004:** Construct correlation matrix and the square root of AVE in the diagonal.

Construct	WRSM	SRSM	KS	KH	KM	Creativity
WRSM	0.640					
SRSM	0.635	0.648				
KS	0.426	0.387	0.742			
KH	−0.292	−0.228	−0.425	0.728		
KM	−0.148	−0.077	−0.334	0.615	0.678	
Creativity	0.390	0.349	0.654	−0.288	−0.316	0.670

**Table 5 behavsci-13-00601-t005:** Fit indices for the estimated model.

Fit Indices	Criteria	Model Results	Fitting
X^2^/df	≤5	2.32	Good fit
RMSEA	≤0.08	0.056	Good fit
CFI	≥0.9	0.89	accepted
TLI	≥0.9	0.87	accepted
SRMR	≤0.08	0.054	Good fit

**Table 6 behavsci-13-00601-t006:** Standardized path coefficients, standard deviation, and 95% CI values.

Hypothesis		β	SD	t-Value	95% CI	Include 0	Result
H1	WRSM → KS	0.368	0.231	1.594	[0.080, 0.773]	No	Supported
H2	SRSM → KS	0.081	0.233	0.345	[−0.396, 0.349]	Yes	Not supported
H3	WRSM → KH	−0.543	0.529	−1.027	[−1.610, −0.077]	No	Supported
H4	SRSM → KH	0.138	0.536	0.257	[−0.343, 1.156]	Yes	Not Supported
H5	WRSM → KM	−0.596	0.596	−0.999	[−1.728, −0.053]	No	Supported
H6	SRSM → KM	0.376	0.602	0.624	[−0.173, 1.521]	Yes	Not Supported
H7	KS → Creativity	0.547	0.034	16.059	[0.480, 0.612]	No	Supported
H8	KH → Creativity	0.043	0.036	1.190	[−0.024, 0.115]	Yes	Not Supported
H9	KM → Creativity	−0.104	0.028	−3.784	[−0.160, −0.053]	No	Supported

## Data Availability

The original contributions presented in the study are included in the article, further inquiries can be directed to the corresponding author.

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
