# Peer review of "Empirical Investigation of How Social Media Usage Enhances Employee Creativity: The Role of Knowledge Management Behavior"

_behavsci, 2023, doi:10.3390/bs13070601_

Round 1
Reviewer 1 Report
Thank you for submitting your manuscript to Behavioral Sciences. It is an interesting read but there are a number of issues that require your attention before this paper can be published. Some are minor and can easily be addressed, others require some further reflection and effort, as indicated below.
Your paper needs some improvement. More specific areas of focus are as follows:
Introduction and Literature Review
The introduction and the literature review can benefit from a more coherent structure. Some sweeping statements and unwarranted arguments make the paper less rigorous and weak.
The general outlook of this section is descriptive and lacks a critical rigour. The author/s out to make use of adequate and recent literature to substantiate their points. Some references like 2004 for something related to social media is not particularly useful especially since the recent advances in IT.
The section on creativity is the weakest! Creativity is only mentioned in passing and there is no adequate build up or link other to the the most commonly researched - idea generation through self assessment.
It would be useful for the authors to delve deeper in the literature related to organisational creativity, both at the individual and the group/organisational level. Work by Michael Mumford, Theresa Amabile, and Roni Reiter-Palmon should offer inspiration.
Method
The sample needs to be described including the inclusion/exclusion criteria used.
Ethical considerations appear to be missing. Moreover, since one IP address was collected the data was not collected anonymously.
Discussion
The discussion is weak since there is hardly any link (only 2) to the literature. Further work is related in this section
Overall
It seems that there were multiple authors due to the different writing styles present in the paper. The paper requires meticulous proof-reading and a better, more coherent outlook.
I hope that the these comments are helpful and that they will help you for your future resubmission.
Best wishes
As it is the paper should not be published. Besides the issues related to the theoretical component the grammar and syntax errors in the paper make it difficult to read. It feels that there were multiple authors due to the different writing styles present in the paper. The paper requires meticulous proof-reading and a better, more coherent outlook.
I
Author Response
Dear reviewer,
We would like to thank you for your valuable comments and kind suggestions regarding the manuscript we submitted. These comments and suggestions directed us to improve the content of our study. We have revised our study substantially, and the revised version of the manuscript incorporates your comments and suggestions. Prior to addressing the comments in a step-by-step manner, we would like to note the following changes in the revised manuscript.
- First, we have revised some statements and arguments and added more contents in Introduction and Literature Review section to improve the logicality and stringency of the paper.
- Second, we have rewritten the section on Employee creativity as the suggestion.
- Third, several sections, such as Method, Discussion has been rewritten based on the comments.
- Fourth, we have added an “Ethics statement” subsection to answer ethics statement questions.
- Finally, more detailed contents about the revision are provided in the revised manuscript.
The details of modification were listed point by point:
[Comment 1]
Introduction and Literature Review
The introduction and the literature review can benefit from a more coherent structure. Some sweeping statements and unwarranted arguments make the paper less rigorous and weak.
The general outlook of this section is descriptive and lacks a critical rigour. The author/s out to make use of adequate and recent literature to substantiate their points. Some references like 2004 for something related to social media is not particularly useful especially since the recent advances in IT.
The section on creativity is the weakest! Creativity is only mentioned in passing and there is no adequate build up or link other to the the most commonly researched - idea generation through self assessment.
It would be useful for the authors to delve deeper in the literature related to organisational creativity, both at the individual and the group/organisational level. Work by Michael Mumford, Theresa Amabile, and Roni Reiter-Palmon should offer inspiration.
[Response] Thank you for your comments. We have carefully addressed all the problems you have mentioned. For the section on Employee creativity, we have rewritten it, the detail statement as below. For more contents of other problems please see detailed revision are provided in the revised manuscript.
Employee creativity: Employee creativity is the creative methods and ideas of employees to solve work problems when they cooperate and interact with others (George & Zhou, 2001), which comes from the knowledge holding and exchange among colleagues (Rhee & Choi, 2017). In other words, employee creativity was identified as employees' use of certain cognitive processes in creative problem solving (Hughes et al., 2018). And the research indicated that in-depth understanding of the cognitive process that facilitate creativity must consider the important factors which developed at multi-level perspective (eg. individual level, team and group level) (Reiter-Palmon et al., 2015). In fact, a large number of studies have been conducted on individual factors such as self-efficacy (Magadley & Birdi, 2012), and organization-level factors as organizational culture (Ogbeibu et al., 2018) and leadership style (Hughes et al., 2018) to explore the impact of various factors on creativity. For example, Zhou et al. (2018) explored the impact of visionary leadership on employee creativity. However, individual creativity is also af-fected by the social network he embedded in, which belongs to the meso level (Sigala & Chalkiti, 2015). Nowadays, the popularity of social media has greatly changed the way of people sharing knowledge, communicating and cooperating (Filo et al., 2015). Reiter-Palmon et al. (2015) indicated that the core cognitive process relevant to crea-tivity including the idea-generation phase (problem identification and construction, information search and encoding, idea and solution generation) and the implementa-tion phase (idea evaluation and selection, implementation planning and monitoring). Social media usage enables individuals to engage in a continuous exchange of ideas in an informal manner with people in same interests. They can share and integrate knowledge from various sources to create new meta-knowledge and to participate in the collective knowledge generation process (Sigala & Chalkiti, 2015). It promotes the development of the core cognitive process including information search and encoding, idea and solution generation, and idea evaluation and selection. Moreover, Mumford et al. (1988) argued that no matter the major or minor creative contributions will re-quire several different knowledges, skills and abilities. Social media usage provides the opportunity to search and gather different knowledge. Meanwhile, there are quite a lot of studies verified the impact of social media usage on employee creativity. For in-stance, Korzynski et al. (2019) shows that social media usage can promote employee creativity through online social knowledge management. Enterprise social media us-age also improve employee creativity with the mediating effect of leader-member ex-change and support for innovation (Wang et al., 2022). Nevertheless, the current stud-ies explored the relationship between social media usage and employee creativity ne-glected how the different purposes of social media usage influences employee creativ-ity in knowledge management perspective. This study will answer the questions by ex-ploring the relationship of social media usage employee creativity and knowledge management behavior to provide insights for future research.
[Comment 2]
Method
The sample needs to be described including the inclusion/exclusion criteria used.
Ethical considerations appear to be missing. Moreover, since one IP address was collected the data was not collected anonymously.
[Response]
Thank you for the detailed comment. We have inserted “Participants” subsection to describe the inclusion/exclusion criteria for the sample and other information. For example, “In the introduction section of the questionnaire, we briefly explained the background and purpose of the survey, emphasizing that participants can voluntarily choose to participate in the questionnaire survey or not. It described “This questionnaire aims to understand the impact of social media usage in daily work, if you are willing to participate in the survey, please select ‘YES’ and complete the survey, if not, please select ‘NO’ and withdraw from the survey." When data is collected, the questionnaire data of some participants who choose not to participate in the survey and fail to complete all the questions were eliminated systematically”.
We also have added an “Ethics statement” subsection. For the question of IP address collected, we only restrict the multiple access of the same IP address systematically and didn’t collect participants’ IP address.
Ethics statement: Ethical review and approval was not required for the study on human participants in accordance with the local legislation and institutional requirements. Written informed consent for participation was not required for this study in accordance with the national legislation and the institutional requirements. Written informed consent was implied via completion of the survey.
[Comment 3]
Discussion
The discussion is weak since there is hardly any link (only 2) to the literature. Further work is related in this section
[Response]
Thank you for your suggestion. We have revised the whole Discussion section and added some literature consents. detailed contents about the revision are provided in the revised manuscript.

Reviewer 2 Report
Thank you for the opportunity to revise this manuscript.
This manuscript is convincingly ambitious, sound, and credible, and has soundness and credible methodology.
- The topic of this manuscript is interesting, and it introduces an innovative aspect. The theme is relevant and has a conceptual domain, the study is a well-conceived, well-crafted, and well-presented paper.
- The methods are appropriate, accurate, and objective for the experiment, and improve the understanding of the reader.
- The analyses are not complex, informative, and appropriate for the experiment.
- The authors recognize the limitations of this research, and in my view, this manuscript has main value and provides evidence that the authors seem to recognize the main innovation of their study.
- Overall, the paper is well-placed to stimulate future research.
- It is clear a substantial amount of work has gone into preparing this manuscript, and it can be reconsidered able to be published and may result in a modest contribution to the literature, but only after revision:
- Please provide one “Participants” subsection and describe more information about the sample
- Please provide the range of age (minimum and maximum) and standard deviation (SD) both in the abstract and participants section,
- Please provide Informed Consent Statement information in the participant's section: The Informed consent was obtained from the subjects involved in the study ? The informed consent was verbal or written?
- Please insert and better describe “Procedures and participants” Subsection.
- Between lines 482 to 500 please provide the Cronbach Alpha for the total scale.
- In table 3 please provide one line with the Total Cronbach Alpha
- Please provide an “Ethics statement” Subsection and provide information about the statement provided for one or more of the other ethics statement questions. Please provide other indicators, e.g. Ethics committee approval and full name of the ethics committee provided. If ethics committee approval was not obtained, a statement specifying this, along with justification of why ethics approval was not obtained or why the ethics committee waived the requirement for approval.
- Please improve limitations and future directions section
- Please improve the Conclusion and discussion section.
Author Response
Dear reviewer,
We would like to thank you for your valuable comments and kind suggestions regarding the manuscript we submitted. These comments and suggestions directed us to improve the content of our study. We have revised our study substantially, and the revised version of the manuscript incorporates your comments and suggestions. Prior to addressing the comments in a step-by-step manner, we would like to note the following changes in the revised manuscript.
- First, we have inserted “Participants” subsection to describe the information about the sample.
- Second, we have added an “Ethics statement” subsection to answer ethics statement questions.
- Third, several sections, such as Data analysis, Discussion have rewritten based on the comments.
- Finally, more detailed contents about the revision are provided in the revised manuscript.
The details of modification were listed point by point:
[Comment]
- It is clear a substantial amount of work has gone into preparing this manuscript, and it can be reconsidered able to be published and may result in a modest contribution to the literature, but only after revision:
- Please provide one “Participants” subsection and describe more information about the sample
- Please provide the range of age (minimum and maximum) and standard deviation (SD) both in the abstract and participants section,
-Please provide Informed Consent Statement information in the participant's section: The Informed consent was obtained from the subjects involved in the study ? The informed consent was verbal or written?
- Please insert and better describe “Procedures and participants” Subsection.
- Between lines 482 to 500 please provide the Cronbach Alpha for the total scale.
- In table 3 please provide one line with the Total Cronbach Alpha
- Please provide an “Ethics statement” Subsection and provide information about the statement provided for one or more of the other ethics statement questions. Please provide other indicators, e.g. Ethics committee approval and full name of the ethics committee provided. If ethics committee approval was not obtained, a statement specifying this, along with justification of why ethics approval was not obtained or why the ethics committee waived the requirement for approval.
- Please improve limitations and future directions section
- Please improve the Conclusion and discussion section.
[Response]
(1) Thank you for the detailed comment. We have inserted “Participants” subsection to describe the Informed Consent Statement information for the sample and other information. The revised section as below.
4.1.2 Participants
During the survey, all participants were informed in written that they could par-ticipate voluntarily. In the introduction section of the questionnaire, we briefly ex-plained the background and purpose of the survey, emphasizing that participants can voluntarily choose to participate in the questionnaire survey or not. It described “This questionnaire aims to understand the impact of social media usage in daily work, if you are willing to participate in the survey, please select ‘YES’ and complete the survey, if not, please select ‘NO’ and withdraw from the survey." When data is collected, the questionnaire data of some participants who choose not to participate in the survey and fail to complete all the questions were eliminated systematically. Data was col-lected only from participants who agreed to participate the survey and completed all questions in the questionnaire. Furthermore, the questionnaire data of participates were removed by manual review based on respondents took less than 2 minutes to fill in the questionnaire and chose the same option for almost all the options.
The employee sample included 53.2 percent male participants. For easy to collect the age of participants, we defined “1= under 25years old,2=26-30 years old,3=31-35 years old, 4= over 35 years old”, the average is 2.21 (SD=0.773). The age of the participants under 25years old (16.5 percent), 26-30 years old (50.6 percent), 31-35 years old (28.0 percent), over 35 years old (4.9 percent). The educational levels of the participants were Associate degree or below (16.7 percent), bachelor (72.5 percent), master/PhD (10.8 percent).
(2) Because we collect participants’ age though age group, we think providing the range of age and standard deviation (SD) in the abstract is inappropriate and we just describe in participants section.
(3) Moreover, we have added an “Ethics statement” subsection and the detailed information as follows.
Ethics statement: Ethical review and approval was not required for the study on human participants in accordance with the local legislation and institutional requirements. Written informed consent for participation was not required for this study in accordance with the national legislation and the institutional requirements. Written informed consent was implied via completion of the survey.
(4) We have added the Cronbach Alpha for the total scale in lines 524 and last line in table 3. Several sections, such as Discussion, Limitation has been rewritten based on the comments. For more contents please see detailed revision are provided in the revised manuscript.

Round 2
Reviewer 1 Report
Dear Authors,
Thank you for resubmitting your paper. Although there are some improvements, the paper still requires a few changes before it is ready for publication.
The section about creativity seems to have been written in a rush. Corrections to the grammar and diction still need to be made
Corrections to the grammar and diction still need to be made
Author Response
Dear reviewer,
We would like to thank you for your detailed comments to help us improve our paper. According to your suggestion on correcting to the grammar and diction, we read through the manuscript and revised some mistakes. Please see detailed revision are provided in the revised manuscript which be marked in bright color.

Reviewer 2 Report
The manuscript was improved with my suggestions
Author Response
Thank you for your review.